# Effect of Training Phase on Physical and Physiological Parameters of Male Powerlifters

**DOI:** 10.3390/sports8080106

**Published:** 2020-07-30

**Authors:** Daniel A. Hackett, Guy C. Wilson, Lachlan Mitchell, Marjan Mosalman Haghighi, Jillian L. Clarke, Yorgi Mavros, Helen O’Connor, Amanda D. Hagstrom, Gary J. Slater, Justin Keogh, Chris McLellan

**Affiliations:** 1Exercise, Health and Performance Faculty Research Group, School of Health Sciences, Faculty of Medicine and Health, The University of Sydney, Lidcombe, NSW 2141, Australia; guy.wilson@sydney.edu.au (G.C.W.); marjan.mosalmanhaghighi@sydney.edu.au (M.M.H.); jillian.clarke@sydney.edu.au (J.L.C.); yorgi.mavros@sydney.edu.au (Y.M.); 2National Nutrition Surveillance Centre, School of Public Health, Physiotherapy and Sport Science, University College Dublin, Belfield, Dublin 4, Ireland; lachlan.mitchell@ucd.ie; 3School of Medical Sciences, Faculty of Medicine, University of New South Wales, Sydney 2052, Australia; m.hagstrom@unsw.edu.au; 4School of Health and Sport Sciences, University of the Sunshine Coast, Sippy Downs, Queensland 4556, Australia; gslater@usc.edu.au; 5Faculty of Health Sciences and Medicine, Bond University, Robina, Gold Coast 4229, Australia; jkeogh@bond.edu.au; 6Sports Performance Research Centre New Zealand, AUT University, Auckland 1142, New Zealand; 7Cluster for Health Improvement, Faculty of Science, Health, Education and Engineering, University of the Sunshine Coast, Sippy Downs, Queensland 4556, Australia; 8Kasturba Medical College, Mangalore, Karnataka 575001, India; 9Manipal Academy of Higher Education, Manipal 576104, Karnataka, India; 10Faculty of Health, Engineering & Sciences, School of Health and Wellbeing, University of Southern Queensland, Ipswich, Queensland 4350, Australia; chris.mclellan@usq.edu.au

**Keywords:** training periodization, strength athletes, muscle power, athletic performance

## Abstract

Longitudinal research on training and dietary practices of natural powerlifters is limited. This study investigated the effect of phases of training on physical and physiological parameters in male natural powerlifters. Nine participants completed testing at two time points: (i) preparatory phase (~3 months prior to a major competition) and (ii) competition phase (1–2 weeks from a major competition). No significant changes between training phases were found for muscle strength and power. A trend for significance was found for decreased muscle endurance of the lower body (−24.4%, *p* = 0.08). A significant increase in leg lean mass was found at the competition phase (2.3%, *p* = 0.04), although no changes for other body composition measures were observed. No change was observed for any health marker except a trend for increased urinary creatinine clearance at the competition phase (12.5%, *p* = 0.08). A significant reduction in training volume for the lower body (−75.0%, *p* = 0.04) and a trend for a decrease in total energy intake (−17.0%, *p* = 0.06) was observed during the competition phase. Despite modifications in training and dietary practices, it appears that muscle performance, body composition, and health status remain relatively stable between training phases in male natural powerlifters.

## 1. Introduction

Powerlifting is a sport that consists of three lifts including the back squat, bench press, and deadlift [1]. The standard competition involves all three lifts. However, there are variations such as single lift competitions, e.g., bench press only. Competitors are divided into body weight, age, and sex categories and have three attempts for each lift to perform a single repetition with a maximal external load while adhering to the judging criteria [1]. In a study investigating the training practices of elite powerlifters, it was reported that 96% of participants used some form of periodization in their training [2]. Generally, a training periodization practice that is commonly incorporated in powerlifting preparation includes time-sensitive manipulation of resistance training volume and intensity to promote peak performance for a targeted competition [3]. Typically, a periodized training program will include two phases: (1) preparatory phase and (2) competition phase [4]. The focus of the preparatory phase is on developing neuromuscular adaptations to increase the loads lifted for the squat, bench press, or deadlift and involves various resistance training methods aimed at improving muscle strength and power [2,4]. During the competition phase, there is a focus on reduction in training workload (i.e., mainly through decreasing training volume), known as ‘tapering’ leading up to a competition in an effort to maximise performance [5,6]. Besides manipulation of resistance training variables between training phases, powerlifters may also change dietary practices to assist with recovery and manipulation of body composition [5]. However, there is a paucity of longitudinal research on powerlifters investigating the effect of training and dietary practices between training phases on muscle performance and body composition.

When compared to the general population, powerlifters have a greater amount of fat-free mass, which has been proposed as the greatest determinant of maximal strength for a powerlifter [7,8,9], although it should be noted that there are a range of other anthropometric variables that could influence powerlifting performance [10]. Since powerlifting competitions have bodyweight categories (with the exception of the super-heavyweights, >120 kg), a desirable body composition for a lifter would be lower in fat mass and higher fat-free mass (i.e., lower fat mass to fat-free mass ratio). Success during a powerlifting competition is not based on aesthetic appearance (e.g., muscularity and low body fat), such is the case for bodybuilders. Therefore, powerlifters may prioritise their diet to gain muscle mass (i.e., via increasing calories) but less attention is given to strict eating for the maintenance of a lean body composition (i.e., lower-body fat). Furthermore, it appears that powerlifters place more emphasis on rapid weight loss through methods such as fluid restriction or water loading prior to competition to “make weight” for competition compared to following a strict dietary plan [11].

Numerous studies have investigated injuries among powerlifters [4,12,13,14] with less attention given to the health status of powerlifters. The associated health risks with anabolic-androgenic steroid usage among athletes such as powerlifters is well documented [15,16,17]. To the best of the authors’ knowledge, there are no studies that have examined comprehensively the health status of natural (without performance-enhancing substances) powerlifters across training phases. The aim of this study was to examine the impact of the preparatory and competition phases of training on physical and physiological parameters in male natural powerlifters. It was hypothesised that muscle strength and power would increase during the competition phase. It was also hypothesised that there would be no change in body composition or health markers between training phases. The findings from this study may provide important information for managing the performance and health of powerlifters across training phases.

## 2. Materials and Methods

### 2.1. Participants

Nine males (median age 36.0 (interquartile range (IQR) 25.5–44.0) years; height 177.0 (173.0–182.0) cm; body mass 87.6 (79.6–101.1) kg; resistance training experience 5.0 (3.0–14.0) years) participated in this study. The participants had 2.0 (1.3–3.0) years of competitive powerlifting experience and resided in Sydney, Australia. Eight of the nine participants competed in all three lifts (bench press, squat and deadlift) and had a Wilks score of 340.2 (316.0–362.9). One participant competed only in the bench press and had a Wilks score of 128.6. Briefly, the Wilks score is a validated method previously used by the International Powerlifting Federation to compare the performance of powerlifters between varying body weight classes [18]. The highest level of competition between participants varied, with four competing at the local (regional) level, two at an international level, one at the state level, and the final participant at a national level. 

An eligibility criteria for the present study was that powerlifters needed to be ‘natural’, thus not using performance-enhancing substances. Information was given to the participants stating possible screening for substances that enhance athletic performance based on the 2013 World Anti-Doping Agency (WADA) List of Prohibited Substances via a urine sample collection and to inform the researchers prior to study commencement if they had taken any prohibited substances during the previous 12 months. However, no actual screening of prohibited substances was conducted. An information statement explaining all procedures and study risks was provided to participants, alongside verbal explanations prior to study participation. Verbal and written consent was provided by participants prior to study commencement. This study was approved by the University of Sydney Human Research Ethics Committee, project number 2015/706. 

### 2.2. Study Design

This study involved conducting various physical and physiological assessments at two time points corresponding to the training phases of male powerlifters that included the (i) preparatory phase (3.0 (2.5–7.0) months prior to a major competition) and (ii) competition phase (1–2 weeks from a major competition). At the two time points, participants were required to visit the laboratory at the University of Sydney, Cumberland Campus. Eight of the nine participants completed testing at the preparatory phase before the competition phase with the exception being the participant that competed in bench press only competitions. Prior to each visit, participants were provided with instructions such as avoiding strenuous exercise at least 24 h prior to the exercise testing and being in an overnight fasted state when visits involved measures where this was a requirement. 

### 2.3. Muscular Performance

Muscular strength, power and endurance were assessed with the chest press and recumbent leg press using Keiser A420 pneumatic resistance training equipment (Keiser Sports Health Equipment, Inc., Fresno, CA, USA). Both these exercises commenced with a concentric contraction prior to the eccentric phase. For each test, the chest press was performed before the leg press. Between each test, participants were monitored for fatigue (via feedback from participants) to ensure that performance would not be negatively affected. Prior to assessing muscular performance, the technique for each exercise was thoroughly explained to participants. Muscular strength for the chest press and leg press was assessed via the one-repetition maximum (1RM) test. Prior to assessing the 1RM, a specific warm-up was performed involving a set of five repetitions at ~50% of perceived 1RM followed by 1–2 sets of 2–3 repetitions at a load corresponding to ~60–80% 1RM. The 1RM protocol involved performing trials of a single repetition of increasing load (~5–10% increments) with 3–5 min rest between attempts. This cycle was continued until the participant was unable to complete a lift, with the 1RM being the heaviest load that was successfully lifted. For one participant, the tester followed a different 1RM protocol at the initial assessment, which has previously been described [19], and the protocol was repeated for the this participant during 1RM testing at the final time point.

Following measurement of 1RM, peak muscle power (W) was assessed at five relative intensities (40%, 50%, 60%, 70%, and 80% 1RM) for the chest press and leg press. Participants were instructed to complete the concentric portion of the repetition as rapidly as possible when verbally cued, then to slowly lower the weight. Three trials of a single repetition were given at each of the five loads specified (3 × 1 × 70% 1RM), separated by a 10–15 s rest period between trials. Participants were asked to take slightly longer rest periods between trials when lifting the higher relative intensities. Power was calculated via the Keiser machines as the average power between 5% and 95% of the concentric phase to eliminate fluctuations at the beginning and end points of motion. Peak power during the concentric phase was calculated via Keiser A400 software which samples force and position (via ultrasonic position transducers) at a rate of 400 times per second. The manufacturer reports that the accuracy of the system to calculate peak power is within 1%. Peak power for the load resulting in the highest power production was recorded. For the chest press, the peak power occurred between 40 and 60% 1RM compared to 50 and 70% 1RM for the leg press. 

After power testing, muscular endurance was assessed by a maximum repetition task at 60% 1RM for the chest press and leg press. Repetitions were performed at a self-determined tempo for the concentric and eccentric phases. The test was ceased if the exercise technique was not appropriate (e.g., failure to perform full range of motion) or if the participant requested to stop the test. Participants were instructed to perform the exercises with the same technique and lifting speeds as per the instructions for the 1RM tests. Isometric handgrip strength of the dominant and non-dominant hands was also assessed using a JAMAR handgrip dynamometer (Sammons Preston, Bolingbrook, IL, USA). Isometric handgrip strength was defined as the peak force achieved from three trials of each hand, with 10–20 s recovery between attempts.

### 2.4. Flexibility

Lower back and hamstring flexibility was indirectly assessed via the sit and reach test. Participants sat on the floor with their legs extended, shoes off, and feet against a sit and reach box. With arms extended in front of the body and hands with fingers extended on top of each other, participants were instructed to bend their trunk forward slowly and progressively (no rebounding) in order to reach the greatest possible distance. To avoid flexing of the knees the tester assisted by applying pressure to the participants’ knees toward the floor. In the maximum flexion position, the participant had to remain still for at least three seconds. The best of three trials was recorded for subsequent statistical analysis.

### 2.5. Body Composition

A whole-body dual energy X-ray absorptiometry scanner (Lunar Prodigy, GE Medical Systems, Madison, WI, USA) was used to measure body composition. Scans were performed under standardised conditions (early morning, overnight fasted, bladder/bowel voided, and standardised body positioning on the scanning bed). Following the scan, in-built analysis software (version 13.60.033; enCORE 2011, GE Healthcare, Madison, WI, USA) allowed the calculation of total and regional (predefined by the software) lean body mass (excluding bone mineral content), fat-free mass and fat mass. Skeletal muscle mass was calculated using an equation described by Kim and colleagues [20]. The equation is as follows:Total Skeletal Muscle Mass =(1.13×appendicular lean mass)−(0.02× age)+(0.61× sex )+0.97
where females are represented with a ‘zero’ and males are represented with a ‘one’ in the equation.

### 2.6. Pulse Wave Velocity and Blood Pressure

Pulse wave velocity (PWV) was assessed using electrocardiogram-gated sequential applanation tonometry (SPT-301, Millar Instruments, Houston, TX, USA) by acquiring waveforms at the brachial and tibial arteries (SphygmoCor 7.1, AtCor, Napereville, IL, USA, www.atcormedical.com). Resting brachial blood pressure was measured in the supine position using an aneroid sphygmomanometer. Participants needed to be lying in the supine position for 10 min prior to the PWV and blood pressure measurements. Participants were measured early morning after an overnight fast. 

### 2.7. Diet and Training Diaries

Participants documented all the food, fluid, and supplements consumed as well as exercise performed (resistance and aerobic training) over a 7-day period at each time point. The diets were analysed using the FoodWorks program (Version 8; Xyris Software, Brisbane, Australia). Total energy intake (kJ per day) and macronutrient intake (g per day) was calculated. Weekly resistance training volume (repetitions × weight × sets) was determined for the upper body (exercises using predominantly upper-body muscles), and lower body (exercises using predominantly lower-body muscles (including the deadlifts)), whereas weekly aerobic training volume was determined by duration (min). 

### 2.8. Blood and Urine Analysis

After an overnight fast (>10 h), participants reported to an accredited commercial laboratory (Douglass Hanly Moir Pty Ltd., Sydney, Australia), where blood samples were collected. Analysis was performed on the same day as that of collection of serum testosterone, calculated free-testosterone, sex hormone-binding globulin (SHBG), insulin-like growth 1 (IGF-1), creatinine, glucose, insulin, lipids (including triglycerides, total cholesterol, high-density lipoprotein cholesterol (HDL-C), and low-density lipoprotein cholesterol (LDL-C)), cortisol, and vitamin D. Participants were also given a container and instructions to collect their urine for a 24 h period. The collection container was then returned to the laboratory (Douglass Hanly Moir Pty Ltd., Sydney, Australia) where the contents were analysed for urine volume, creatinine excretion and creatinine clearance corrected for surface area. 

### 2.9. Statistical Analysis

Statistical analyses were performed using SPSS version 24.0 for Windows (IBM Corp., Armonk, NY, USA). Data were inspected visually and statistically for normality using the Kolmogorov–Smirnov test. Given the small subject sample size and lack of normal data distribution, the Wilcoxon signed-rank test was used for preparatory and competition phase comparisons. Data are presented as median with interquartile range (IQR). Median percent change was calculated using the following formula: (competition phase minus preparatory phase median) divided by (preparatory phase median) multiplied by 100. Significance level was set at *p* < 0.05 and trends declared at *p* = 0.05–0.10.

## 3. Results

### 3.1. Exercise Performance and Body Composition between Training Phases

Results for exercise performance and body composition are presented in Table 1. No significant change was found for 1RM and peak power for both the chest press and leg press. A trend for significance was found for leg press muscle endurance (Z = −1.78, *p* = 0.08), with a 24.4% decrease in repetitions performed. No significant change was found for chest press muscular endurance (median change = 10.0%, *p* > 0.05). It should be noted that two participants chose not to complete the leg press measures due to concerns about injury and negative impact on competition performance. There were also no statistically significant changes found for handgrip strength as well as sit and reach performance. A significant increase in leg lean mass was found (Z = −2.1, *p* = 0.04) with a median change of 2.3%. There were no significant changes for all other body composition measures. 

### 3.2. Physiological Health Parameters between Training Phases

Table 2 displays the results of the arterial, blood and urine parameters during the preparatory and competition phases. Note that one participant was unable to attend the laboratory to have blood and urine collected and analysed. No significant changes were found for systolic and diastolic blood pressure and pulse wave velocity or for any anabolic hormone measures (testosterone, free testosterone, SHBG, IGF-1, free androgen index). There were no significant changes found for creatinine, glucose, insulin, lipids, cortisol, vitamin D, urine volume, and creatinine excretion. There was a trend for an increase in creatinine clearance (corrected for surface area) (Z = −1.78, *p* = 0.08), with a median change of 12.5%. The median for creatinine excretion during the preparatory phase was 21.5 (20.0–23.5) mmol/day, which exceeded the reference range of 8.8–18.0 mmol/day, and remained above the reference range at the competition phase (23.1 (20.0–25.6) mmol/day). All other blood and urine parameters were within the reference ranges for apparently healthy adults. 

### 3.3. Diet and Exercise between Training Phases

Diet and training results are presented in Table 3. Note there were two participants that did not return their diet and exercise diaries, and therefore data from a total of seven participants were completed for these parameters. There were statistical trends found for a decrease in total energy intake (median change = 17.0%, Z = −1.86, *p* = 0.06), increase in fat intake (median change = 2.4, Z = −1.69, *p* = 0.09), and increase in carbohydrate intake (median change = 8.5%, Z = −1.86, *p* = 0.06). There was no significant change found for protein intake. Resistance training volume for the upper body did not significantly change between phases despite a median change of −54.4%. In contrast, lower-body resistance volume significantly decreased (median change = −75.0%, Z = −2.03, *p* = 0.04). However, when the participant that competed in bench press only was removed from the lower-body volume analysis (since he reported no lower-body exercises at the competitive phase), a trend was found (median change = −68.0%, Z = −1.78, *p* = 0.08). Only two participants performed aerobic exercise in the preparatory phase (total volume = 22 and 82 min/week), with the same two participants also performing aerobic exercise in the competition phase (total volume = 10 and 105 min/week). 

## 4. Discussion

The purpose of this study was to examine the impact of preparatory and competition phases of training on physical and physiological parameters in male powerlifters. Contrary to the original hypothesis, greater muscle strength and power were not observed during the competition phase. However, there was a trend for a reduction in lower-body muscle endurance performance (i.e., decreased number of repetitions) during the competition phase. Leg lean mass was greater during the competition phase but no other body composition changes were observed during this phase, which was in agreement with the original hypothesis. In support of the original hypothesis, the majority of clinical markers of health status (i.e., arterial, blood and urine parameters) did not differ between training phases. The only potential exception was a trend for increased urinary creatinine clearance at the competition phase. There was a trend for decreased total energy intake during the competition phase. Trends were also observed for increased intake of fats and carbohydrates during the competition phase, but there was no difference for protein intake between training phases. A reduced resistance training volume was observed for the lower body during the competition phase but no difference in training volume was found between training phases for the upper body. The overall results suggest that physical and physiological parameters of male powerlifters remain relatively stable between preparatory and competition phases of training despite manipulation of training and dietary practices. However, some caution is warranted when interpreting the study findings due to the small size and limited testing (i.e., conducted at only two time points). 

To maximise competitive performance, a prior taper period is usually followed which is generally identified by decreases in training volume to aid recovery [5]. In the present study, training volume was reduced by 54% for the upper body and 75% for the lower body ≤ 2 weeks from competition. Although, only for the lower body was training volume found to be significantly different to the preparatory phase. This is contrary to previous tapering practices of powerlifters where self-reported reductions in training volume for the upper and lower body are the same (i.e., identified by reference to the three lifts) [5]. An explanation for the lack of improvement in upper-body strength and power may be due to training volume during the competition phase not being significantly reduced. However, since there was adequate tapering for the lower body (and trend when bench press only participant was excluded), it is surprising that improvements in muscle performance for this body region were not observed. 

The absence of improvement in lower-body muscle strength and power found in the present study is further complicated by the trend for a reduction in lower-body muscle endurance during the competition phase. Following a tapering period in strength athletes improvements of 2–3% have been found for the bench press and squat [21]. These strength improvements may be attributed to a reversal in neuromuscular fatigue [22] and general recovery identified by reduced circulating markers of muscle damage [23]. Even though participants were encouraged to perform to the best of their abilities, they may have been less motivated during the competition phase tests due to fears of adverse effects (fatigue, soreness) or events (e.g., injury) prior to competition. Further, it is possible that the reduction in training volume and the tendency to perform sets with very high loads and low number repetitions may have contributed to a reduced strength endurance capacity. The results for lower-body muscle performance are unlikely to be influenced by the participant that competed only in the bench press since at the competition phase, he had < 1.0% increase in strength, 16.0% increase in power, and 10% decrease in muscle endurance. 

There was minimal change in the body composition of participants between the preparatory and competition phases. Body mass and fat-free mass have been proposed as two main determinants of powerlifting performance [7,8]. Ideally a low fat mass to fat-free mass ratio would be advantageous for a powerlifter, since competitions have bodyweight categories (except for super-heavyweights, >120 kg). Further, if there is a tied event (i.e., equal total loads lifted), the winner will be the competitor with the lighter body mass at weigh in [11]. Additionally comparison across powerlifters and weight categories can be performed through total loads lifted adjusted to body mass (e.g., Wilks score) [18]. In the present study, total body mass, fat-free mass and fat mass did not significantly change between training phases despite there being a 17% decrease in caloric intake during the competition phase. This may indicate that participants rely on rapid weight loss methods since it was recently reported that approximately 86% of competitive powerlifters engage in this practice [11]. Despite the common practice of large rapid weight loss strategies prior to weigh in that traditionally occurs < 2 h prior to competition in most drug-tested powerlifting federations [1], the concomitant risk of a reduction in performance due to time constraints associated with re-hydration and refueling likely outweigh the perceived benefit of these pre-event tactics. The magnitude of rapid weight loss in powerlifters has been reported to be ~3% of body mass [11]. While it appears that body composition is quite stable across the training phases, there was an increase in leg lean mass observed during the competition phase. Participants increased leg lean mass by 2.3%. However, this does not seem to be meaningful, since lower-body muscle strength and power showed no improvement at the competition phase.

Elevated resting blood pressure has been reported in strength trained athletes (powerlifting, bodybuilding, and Olympic weightlifting) [24]. Resting blood pressure ≥ 130/80 mmHg (previously ≥ 140/90 mmHg) is defined as hypertension [25] and is considered a risk factor for stroke, heart attack, kidney failure and congestive heart failure [26]. In a group of natural powerlifters, it was shown that resting blood pressure was significantly higher compared to a group of endurance runners (130/82 mmHg versus 116/72 mmHg, respectively) [27]. In the present study, the resting blood pressure of the participants was considered healthy since the readings were less than 120/80 mmHg [25] and did not change between training phases. High-intensity resistance training (as is routinely performed by powerlifters) has been shown to increase arterial stiffness [28], perhaps due to increases in sympathetic nervous activity which stimulates vasoconstriction [29] and increased oxidative stress, leading to endothelial dysfunction [30]. Arterial stiffness is a predictor of coronary heart disease risk [31], with the normal (healthy) median values being < 7 m/s [32]. It was therefore interesting to observe that participants in this study had healthy arterial stiffness median values during the preparatory phase and that this did not change during competition.

Hormone, lipid and glucose values were within the normal reference range for the participants and there was no change observed between training phases. This finding is consistent with other health outcomes and suggests that the participants in this study were relatively healthy with their health status maintained throughout the training phases. However, the participants weighed less than 90 kg and had < 20% body fat, so it is likely that a less favourable health status would have been found if the participants were heavier with greater body fat due to the well-established associated health risks [33]. The only abnormal finding for health status was higher urinary creatinine excretion compared to reference values. This may be explained by increased fat-free mass and engagement of intense exercise maintained throughout both training phases which is shown to elevate urinary creatinine excretion [34]. When creatinine clearance was corrected for body surface area there was a trend for greater levels at the competition phase, although these were within the reference values. Besides the other factors mentioned above, it has been proposed that a high dietary intake of protein can increase creatinine excretion and clearance [35]. The participants daily protein intake was > 2 g/kg/body weight per day which is more than double the recommendation for adults [17]. Although the protein recommendation for strength trained athletes is 1.5–2.0 g/kg bodyweight per day [36], it is common for strength and power athletes’ protein intake to be well over 2 g/kg bodyweight per day [37,38]. While there is little evidence on the longitudinal health effects of a high-protein diet, there appears to be concerns that daily protein intake above 1.5 g/kg bodyweight may increase the risk of chronic kidney disease [39].

There are several limitations that need to be acknowledged when interpreting the findings from the present study. This study only measured outcomes at two time points and if testing occurred over a greater number of time points, there would have been a better representation of the effect of training phase on physical and physiological parameters. The median time period prior to competition for the preparatory phase was only 3 months, which may not have provided enough time to compare the ‘true’ effects of training and dietary practices on the measures. Additionally, there was large variation among participants for the time period between the preparatory and competition phases (ranging from 2 to 10 months). This could have influenced physical and physiological results. Specifically, there would be an increased potential for changes with greater time between the testing periods. An explanation for the large variation was due to some participants needing to select another competition to target weeks or months after the initial testing because of work or personal commitments. It is possible that muscle performance testing at the competition phase was influenced by apprehension, although as previously stated participants were verbally encouraged to perform to the best of their abilities for all physical tests. The generalisability of the study is reduced due to the small sample size and there were certain measures that 1–2 participants did not complete, e.g., blood and urine analysis, training and diet diaries. Therefore, it is possible that type 2 errors (false negatives) may have occurred. However, the findings from this study do provide insightful information and due to the paucity of research exploring this topic, the study may assist in the future development of adequately powered and well-designed study designs. 

## 5. Conclusions

The findings from this study show that natural male powerlifters maintain relatively stable muscle performance, body composition, and good health status between the preparatory and competition phases of training. This is despite modifications to their practices in which training volume is reduce (i.e., tapering) and caloric restriction is followed during the competition phase. Training status might have influenced the muscle performance results of the participants since it has been shown that improvement in muscle performance is more difficult in athletes that have reached close to their maximum physical capabilities (i.e., ‘ceiling effect’). This is referred to as the law of diminishing returns and states that less improvements from training occur as strength levels and training ages increase. For example, it was reported that well-trained rugby league players increased their peak power but only by 5% over a 4 year period [40]. Another possible factor that may have influenced the muscle performance results is the large variation between testing periods among participants. For participants that had less time between the preparatory and competition phases, improvement in muscle performance may have been limited, hence training was merely for the maintenance of their performances. Due to the small sample size, further research is required to substantiate the current study’s findings.

## Figures and Tables

**Table 1 sports-08-00106-t001:** Exercise performance and body composition during the preparatory and competition phases.

Parameter	Preparatory Phase	Competition Phase	% Change	Z	*p* Value
CP 1RM (N)	1025.0 (925.0–1112.5)	1005.0 (935.0–1150.0)	−2.0	−1.40	0.16
CP peak power (W)	862.0 (811.0–1079.5)	890.0 (769.5–1094.5)	3.2	−0.89	0.37
CP endurance (rep)	20.0 (18.5–21.0)	22.0 (18.5–24.0)	10.0	−1.62	0.11
LP 1RM (N) ^b^	4025.0 (3804.8–4162.5)	4000.0 (3625.0–4250.0)	−0.6	−1.19	0.24
LP peak power (W) ^a^	2561.0 (2008.8–2862.3)	2332.8 (1891.0–3338.0)	−8.9	−0.68	0.50
LP endurance (rep) ^a^	41.0 (28.5–43.8)	31.0 (25.0–33.0)	−24.4	−1.78	0.08 ^†^
Handgrip—right (kg) ^b^	51.0 (47.0–52.0)	52.3 (43.0–62.0)	2.5	−0.85	0.40
Handgrip—left (kg) ^b^	50.0 (42.3–59.5)	50.0 (47.0–58.0)	0.0	−0.93	0.35
Sit and reach (cm)	31.5 (21.3–34.3)	30.5 (20.5–36.8)	−3.2	−0.68	0.50
Total mass (kg)	88.3 (81.6–101.7)	86.4 (80.1–101.6)	−2.2	−1.37	0.17
Fat mass (kg)	15.7 (9.0–23.9)	16.6 (8.4–22.8)	5.7	−0.89	0.37
Lean mass (kg)	70.3 (63.5–74.0)	68.3 (64.3–73.9)	−2.8	−0.77	0.44
FFM (kg)	74.3 (67.0–77.8)	72.4 (67.6–77.8)	−2.7	−0.65	0.52
SMM (kg)	37.0 (34.1–41.5)	36.7 (34.9–42.2)	−0.6	−1.36	0.17
Body fat (%)	17.6 (12.6–25.0)	19.5 (11.7–24.0)	10.8	−0.83	0.41
Android fat (kg)	1.4 (0.8–2.5)	1.5 (0.7–2.4)	7.7	−1.01	0.31
Arms lean mass (kg)	10.4 (9.1–11.0)	9.7 (9.1–10.8)	−6.7	−0.77	0.44
Legs lean mass (kg)	21.6 (20.0–24.8)	22.1 (20.3–25.6)	2.3	−2.07	0.04^*^

* Significant at *p* < 0.05; ^†^ trend to significance at *p* = 0.05–0.10; ^a^ based on *n* = 7; ^b^ based on *n* = 8. CP = chest press; N = Newtons; W = watts; rep = repetitions; 1RM = one-repetition maximum; LP = leg press; FFM = fat-free mass; FFMI = fat-free mass index; SMM = skeletal muscle mass.

**Table 2 sports-08-00106-t002:** Arterial, blood and urine parameters during the preparatory and competition phases (*n* = 8).

Parameter	Reference Range ^a^	Preparatory Phase	Competition Phase	% Change	Z	*p* Value
SBP (mmHg)		115.0 (109.5–121.0)	110.0 (10.5.0–114.3)	−4.3	−1.54	0.13
DBP (mmHg)		71.0 (64.5–78.5)	66.5 (61.8–71.0)	−6.3	−0.51	0.61
PWV (m/s)		6.5 (5.9–7.5)	6.6 (5.9–7.1)	1.5	−1.55	0.12
Testosterone (nmol/L)	11.5–32.0	20.8 (18.9–22.5)	20.2 (18.3–21.0)	−2.9	−0.77	0.44
Free testosterone (pmol/L)	260–740	350.0 (309.5–444.5)	350.0 (311.5–414.3)	0	−0.77	0.44
SHBG (nmol/L)	15–50	44.0 (33.5–53.5)	44.0 (35.3–48.0)	0	−0.43	0.67
Free androgen index (%)	15–100	42.8 (38.5–63.4)	45.0 (37.1–59.5)	5.1	−0.98	0.33
IGF-1 (nmol/L)	21–68	27.0 (24.5–33.0)	27.5 (27.0–31.3)	1.9	−1.33	0.18
Creatinine (umol/L)	60–110	80.0 (75.0–117.5)	80.0 (76.3–88.8)	0	−0.43	0.67
Glucose (mmol/L)	3.6–6.0	4.5 (4.3–4.8)	4.7 (4.2–5.0)	4.4	−0.34	0.73
Insulin (mU/L)	0–20	4.0 (3.5–8.0)	5.0 (4.0–6.0)	25	−0.55	0.58
Triglycerides (mmol/L)	0.5–1.7	0.6 (0.5–0.8)	0.7 (0.4–1.2)	16.7	−0.51	0.61
Total cholesterol (mmol/L)	3.9–5.5	4.7 (4.2–5.5)	5.0 (3.8–5.3)	6.4	−1.47	0.14
HDL (mmol/L)	0.8–1.5	1.4 (1.4–1.7)	1.3 (1.2–1.6)	−7.2	−0.94	0.35
LDL (mmol/L)	1.7–3.5	3.1 (2.6–3.5)	3.0 (2.2–3.4)	−3.2	−1.20	0.23
Cortisol (nmol/L)	138–650	279.0 (228.0–412.5)	369.5 (274.3–406.0)	32.4	−0.42	0.67
Vitamin D (nmol/L)	50–140	85.0 (63.5–133.5)	94.5 (58.5–98.5)	11.2	−0.34	0.74
Urine volume (mL)		3220.0 (1930.0–5065.0)	3225.0 (2530.0–4785.0)	0.2	−0.28	0.78
Creatinine excretion (mmol/day)	8.8–18.0	21.5 (20.0–23.5)	23.1 (20.0–25.6)	7.4	−1.01	0.31
Creatinine clearance (mL/s) ^b^	1.10–3.20	2.4 (1.9–2.8)	2.7 (2.3–3.3)	12.5	−1.78	0.08 ^†^

^a^ Reference range for apparently healthy adults provided by Douglass Hanly Moir Pty Ltd., Sydney, Australia; ^b^ Corrected for surface area; ^†^ trend to significance at *p* = 0.05–0.10. SBP = systolic blood pressure; DBP = diastolic blood pressure; PWV = pulse wave velocity; SHBG = sex hormone-binding globulin; IGF-1 = insulin-like growth 1; HDL = high-density lipoprotein cholesterol; LDL = low-density lipoprotein cholesterol.

**Table 3 sports-08-00106-t003:** Diet and exercise training during the preparatory and competition phases (*n* = 7).

	Preparatory Phase	Competition Phase	% Change	Z	*p* Value
Total energy (kJ)	14,639.7 (10,749.8–16,275.3)	12,156.8 (10,049.5–14,720.4)	−17.0	−1.86	0.06 ^†^
Protein (g)	229.1 (169.4–245.2)	210.6 (137.7–216.8)	−8.1	−1.52	0.13
Fat (g)	89.9 (86.8–139.6)	92.1 (82.4–111.5)	2.4	−1.69	0.09 ^†^
Carbohydrates (g)	276.2 (252.8–338.3)	299.8 (219.2–308.2)	8.5	−1.86	0.06 ^†^
UB RT volume (kg)	22,732.5 (7290.0–30,123.5)	10,377.0 (3560.0–19,320.0)	−54.4	−1.35	0.18
LB RT volume (kg)	16,662.5 (9975.0–27,170.0)	4180.0 (1312.5–10,113.0)	−75.0	−2.03	0.04 *

* Significant at *p* < 0.05; ^†^ trend to significance at *p* = 0.05–0.10. UB RT = upper-body resistance training; LB RT = lower-body resistance training.

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
