# Peer review of "Effect of Training Phase on Physical and Physiological Parameters of Male Powerlifters"

_sports, 2020, doi:10.3390/sports8080106_

Round 1

Reviewer 1 Report

This is a clear and well written study dealing with an important issue also relevant for other sports in which making weight before competition without losing strength/ power, is common practice. Unfortunately, the impact of the present study will not be very high, this is due to a number of limitations. These limitations (the limited sample size being an important one) are carefully discussed by the authors. In addition, the present study also shows how difficult it is to get a complete (and reliable) set of data working with athletes who  (understandably) prioritize their sports above scientific testing. So I agree with the authors: However, the366 findings from this study do provide insightful information and due to the paucity of research367 exploring this topic the study may assist in the future development of adequately powered and well368 designed study designs.

An eligibility criterion for the present study was that powerlifters needed to be ‘natural’, thus not using performance enhancing substances. Information was given to the participants stating possible screening for substances

Please clarify in the manuscript : did actual screening on drugs take place, or were the participants only told that testing could occur?

109 phase (3.0 [2.5-7.0] I take it, that this is the range, but why is this so broad, surely there are practical issues , but he variation is very large

Regarding the power testing 5 weights x 3 attempts with 10-15 sec rest in between (or slightly longer) seems a very short rest period to me even for these specialists. In addition, it always has been my understanding that despite the name of the sport, power isn’t the real issue, since movement velocity is low (the emphasize is on strength). Why were the athletes tested for power? The 1 RM seems the crucial/most specific test to me. Did these athletes perform power like exercises in training, in other words: were they used to executions with the attempt to accelerate as fast as possible?

Provide more details about how exactly peak power was determined: Power was calculated via the Keiser machines etc. isn’t clear enough. Usually, considering the linear force-velocity relationships during multi joint exercises, peak power is obtained when the exercise is conducted with a weight that is about 50% 1 RM. Were these relations constructed or was peak power calculated separately for each of the loads (40-80 % 1RM) and was the value reported for the load at which the highest power production was recorded?

Was the duty cycle fixed during endurance testing? Or were athletes allowed to choose their own pace? Please add this information

I wonder what  the isometric handgrip strength test adds to the study please provide the rational 

Similarly: it would surprise me if the flexibility test would add something significant, what was the rational to include this?

Participants174 were measured early morning after an overnight fast. Please provide more details about the exact procedure e.g. how long did the subjects lay down before the measurement took place

Figure 1 is redundant, since the data already are presented in the table 1

Discussion

However, some caution is warranted when interpreting the study277 findings due to the small size and limited testing (i.e. conducted at only two time points).

I fully agree

In the present study, the resting blood pressure of328 the participants was considered healthy since the readings was less than 120/80 mmHg [25] and did329 not change between training phases.

May this have been die to the fact that you measured the participants in supine (and not seated) position?

Author Response

We thank you for your constructive comments which have enabled us to improve the manuscript. Please find below a point-by-point response to all the comments raised. 

 Comment 1: An eligibility criterion for the present study was that powerlifters needed to be ‘natural’, thus not using performance enhancing substances. Information was given to the participants stating possible screening for substances

Please clarify in the manuscript: did actual screening on drugs take place, or were the participants only told that testing could occur?

Response: The participants were only told this information in an attempt to deter people using performance enhancing drugs from participating. So no actual screening took place. The following has been added to the Methods (Line 102):

“However, no actual screening of prohibited substances was conducted.”

Comment 2: 109 phase (3.0 [2.5-7.0] I take it, that this is the range, but why is this so broad, surely there are practical issues , but he variation is very large

Response: The reason for the broad range was that some participants had to change competitions that were targeting (due to work, personal issue etc.) following the initial testing session. This definitely may have influenced results and the following information was added to the Discussion (Lines 360-366):

“Additionally, there was large variation among participants for the time period between the preparatory and competition phases (ranging from 2-10 months). This could have influenced physical and physiological results. Specifically, there would be an increased potential for changes with greater time between the testing periods. An explanation for the large variation was due to some participants needing to select another competition to target weeks or months after the initial testing because of work or personal commitments.”

Comment 3: Regarding the power testing 5 weights x 3 attempts with 10-15 sec rest in between (or slightly longer) seems a very short rest period to me even for these specialists.

Response: The recovery between attempts did not appear to be too short for the lighter loads because best performances were very similar among all attempts (approximately less than 5%), but as stated, “slightly longer rest periods between trials when lifting the higher relative intensities” – which in some cases could have been 1 minute. But in any case, the single highest peak power produced out of any of the loads was used for data analysis for the chest press and leg press.

Comment 4: In addition, it always has been my understanding that despite the name of the sport, power isn’t the real issue, since movement velocity is low (the emphasize is on strength). Why were the athletes tested for power? The 1 RM seems the crucial/most specific test to me. Did these athletes perform power like exercises in training, in other words: were they used to executions with the attempt to accelerate as fast as possible?

Response: The intention to move a load quickly (which is power training) is heavily emphasised by powerlifters during training and competition. You are correct that velocity is low during competition (which is related to the load used), hence peak power is not an important factor influencing success. Muscle power is affected more by strength than changes in velocity, but it was of interest to assess whether peak power would change throughout the training cycles since powerlifters lift with the intention to move loads as quickly as possible.

Comment 5: Provide more details about how exactly peak power was determined: Power was calculated via the Keiser machines etc. isn’t clear enough. Usually, considering the linear force-velocity relationships during multi joint exercises, peak power is obtained when the exercise is conducted with a weight that is about 50% 1 RM. Were these relations constructed or was peak power calculated separately for each of the loads (40-80 % 1RM) and was the value reported for the load at which the highest power production was recorded?

Response: Thank you for bringing up this question to help improve our manuscript. The following has been added to the Methods (Lines 140-145).

“Peak power during the concentric phase was calculated via Keiser A400 software which samples force and position (via ultrasonic position transducers) at a rate of 400 times per second. The manufacturer reports that the accuracy of the system to calculate peak power is within 1%. Peak power for the load resulting in the highest power production was recorded. For the chest press the peak power occurred between 40-60% 1RM compared to 50-70% 1RM for the leg press.”

Comment 6: Was the duty cycle fixed during endurance testing? Or were athletes allowed to choose their own pace? Please add this information

 Response: Participants were allowed to choose their own pace. The following has been added to the Methods (Lines 148-149).

“Repetitions were performed at a self-determined tempo for the concentric and eccentric phases.”

Comment 7: I wonder what the isometric handgrip strength test adds to the study please provide the rational.

Response: Handgrip is a commonly used measure for general strength in various populations. This was included since researchers in the future might be interested in extracting this data for either a systematic review or to reference.

Comment 8: Similarly: it would surprise me if the flexibility test would add something significant, what was the rational to include this?

Response: Flexibility is a measure that is commonly used in general fitness testing batteries and this measure has been shown to be associated with numerous indicators of health. Both handgrip and flexibility results from the powerlifters might be of interest to researchers in the future.

Comment 9: Participants were measured early morning after an overnight fast. Please provide more details about the exact procedure e.g. how long did the subjects lay down before the measurement took place.

 Response: The following information has been added to the Methods (Lines 180-182).

“Participants needed to be lying in the supine position for 10 minutes prior to the PWV and blood pressure measurements.”

Comment 10: Figure 1 is redundant, since the data already are presented in the table 1

Response: We agree and have decided to delete Figure 1.

Comment 11: In the present study, the resting blood pressure of the participants was considered healthy since the readings was less than 120/80 mmHg [25] and did not change between training phases.

May this have been due to the fact that you measured the participants in supine (and not seated) position?

Response: Supine blood pressure ≥ 130/80 mm Hg is used as specific and sensitive threshold for diagnosis of hypertension. Please see this reference:

  KrzesiÅ„ski, P., StaÅ„czyk, A., Gielerak, G., Piotrowicz, K., Banak, M., & Wójcik, A. (2016). The diagnostic value of supine blood pressure in hypertension. Archives of medical science : AMS12(2), 310–318. https://doi.org/10.5114/aoms.2016.59256

Reviewer 2 Report

line 69 :  powerlifters may prioritise their diet to gain muscle mass 70 (i.e. via increasing calories) maybe proteins ?

give please an explanation for this change (table 1): Arms lean mass (kg) 10.4 (9.1-11.0) 9.7 (9.1-10.8) -6.7 -0.77 0.44

The Conclusion must be expanded. They are too short. Maybe an explanation for no change resides on the fact that the athletes has reached their maximum physical capacities, thus the training is a merely maintenance of this top. In the literature there are some references about this , and should be included in the conclusions. I the conclusion, possible theorization and rationale for the results should be given in extent.

Author Response

We thank you for your constructive comments which have enabled us to improve the manuscript. Please find below a point-by-point response to all the comments raised. 

 Comment 1: line 69:  powerlifters may prioritise their diet to gain muscle mass  (i.e. via increasing calories) maybe proteins ?

Response: To gain muscle and weight total calories are increased for powerlifters. See the recommendation from the following case study:

Oliver, J.M., Mardock, M.A., Biehl, A.J. et al. Macronutrient intake in Collegiate powerlifters participating in off season training. J Int Soc Sports Nutr 7, P8 (2010). https://doi.org/10.1186/1550-2783-7-S1-P8

Comment 2: Give please an explanation for this change (table 1): Arms lean mass (kg) 10.4 (9.1-11.0) 9.7 (9.1-10.8) -6.7 -0.77 0.44).

Response: Increasing muscle mass is not a main objective for a powerlifter. The reason it decreased, although this was not significant, is probably a combination of decreased calories and reduced training volume. This is purely speculative, but since the change in upper lean mass was not significant we do not believe a discussion of this topic is needed.

Comment 3: The Conclusion must be expanded. They are too short. Maybe an explanation for no change resides on the fact that the athletes has reached their maximum physical capacities, thus the training is a merely maintenance of this top. In the literature there are some references about this, and should be included in the conclusions. I the conclusion, possible theorization and rationale for the results should be given in extent.

Response: Thank you for providing this useful suggestion to help improve our manuscript. The following information has been added to the Conclusion (Lines 385-395).

“Training status might have influenced the muscle performance results of the participants since it has been shown that improvement in muscle performance is more difficult in athletes that have reached close to their maximum physical capabilities (i.e. ‘ceiling effect’). This is referred to as the law of diminishing returns and states that less improvements from training occur as strength levels and training age increases. For example, it was reported that well-trained rugby league players increased their peak power but only 5% over a 4-year period [40]. Another possible factor that may have influenced the muscle performance results is the large variation between testing periods among participants. For participants that had less time between the preparatory and competition phases, improvement in muscle performance may have been limited, hence training was merely for the maintenance of their performances.”

Round 2

Reviewer 1 Report

I have no further comments.

Reviewer 2 Report

The changes added in the revision are acceptable, the paper is now suitable for publication.